# SPARSE TOKEN TRANSFORMERS WITH ATTENTION BACK TRACKING

**Heejun Lee**[1,2]   **Minki Kang**[1,3]   **Youngwan Lee**[1,4]   **Sung Ju Hwang**[1]

KAIST[1], DeepAuto.ai[2,*] AITRICS[3], ETRI[4]

{ainl, zzxc1133}@kaist.ac.kr  yw.lee@etri.re.kr
sjhwang82@kaist.ac.kr

## ABSTRACT

Despite the success of Transformers in various applications from text, vision, and speech domains, they are yet to become standard architectures for mobile and edge device applications due to their heavy memory and computational requirements. While there exist many different approaches to reduce the complexities of the Transformers, such as the pruning of the weights/attentions/tokens, quantization, and distillation, we focus on token pruning, which reduces not only the complexity of the attention operations, but also the linear layers, which have non-negligible computational costs. However, previous token pruning approaches often remove tokens during the feed-forward stage without consideration of their impact on later layers' attentions, which has a potential risk of dropping out important tokens for the given task. To tackle this issue, we propose an attention back-tracking method that tracks the importance of each attention in a Transformer architecture from the outputs to the inputs, to preserve the tokens that have a large impact on the final predictions. We experimentally validate the effectiveness of the method on both NLP and CV benchmarks, using Transformer architectures for both domains, and the results show that the proposed attention back-tracking allows the model to better retain the full models' performance even at high sparsity rates, significantly outperforming all baselines. Qualitative analysis of the examples further shows that our method does preserve semantically meaningful tokens.

## 1 INTRODUCTION

Transformers have achieved huge success in various application domains such as natural language processing (NLP) and computer vision (CV), obtaining state-of-the-art performances on a variety of tasks, and are now considered the de facto standard architectures for a number of domains. However, Transformers require over a few tera-flops per entry (Devlin et al., 2019; Dosovitskiy et al., 2021b) to compute, which is orders of magnitude larger than the computational cost for previous CNN and RNN architectures (Tan & Le, 2019; Bahdanau et al., 2015). To reduce such computational burdens of Transformer models, previous works explored model compression methods such as distillation (Wang et al., 2020; Jiao et al., 2020), quantization (Han et al., 2015; Frankle & Carbin, 2018), and pruning (Zaheer et al., 2020; Guo et al., 2019; Kim et al., 2022; Rao et al., 2021; Wang et al., 2021). Pruning approaches for Transformers mostly aim to remove unnecessary model weights (Han et al., 2015; Frankle & Carbin, 2018), or the attentions (Zaheer et al., 2020), which could achieve a linear reduction in complexity. Token pruning can reduce the complexity of attentions and fully connected layers simultaneously by removing less relevant tokens for the target task (Kim et al., 2022; Goyal et al., 2020; Kong et al., 2021). How can we decide which tokens to prune then? Previous studies either compute the importance score of each input token as the average attention scores (Goyal et al., 2020; Kim et al., 2022) (Figure 1), or learn the importance score of each token with an additional neural network (NN) at each layer (Rao et al., 2021; Kong et al., 2021).

However, previous token pruning methods have the following problem: they prune the tokens in the input sequence without explicitly evaluating the importance of each token on the **final sequence representation** and **prediction tasks**. This is because all existing works (Kim et al., 2022; Wang

---

*DeepAuto.ai, Seoul, South Korea

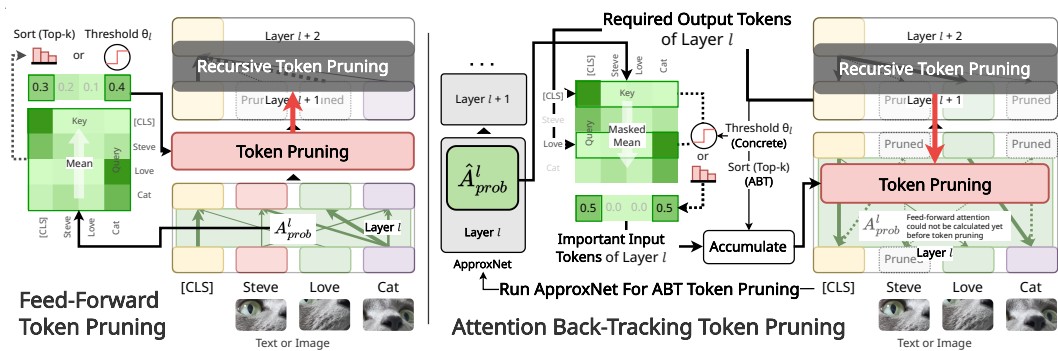

Figure 1: **Concepts.** Comparison of token pruning method between feed-forward and attention back-tracking. The thickness of arrows represents the weight of attention probability. The left side is token pruning in a feed-forward pass. The right side is token pruning with attention back-tracking.

et al., 2021; Goyal et al., 2020; Rao et al., 2021; Kong et al., 2021) compute the importance score of an input token at each attention layer while doing the forward pass from the input to the final output (feed-forward token pruning, Figure 1). Thus, they may prune out important tokens at earlier layers that are important for the final representation as well as the task loss. For example, in Figure 1, the token Love in the layer $l + 1$ is pruned by the feed-forward method, although the token has a high attention probability in the representation at the final layer $l + 2$ that is used by the classification task.

To tackle such a limitation of conventional feed-forward token pruning methods, we propose an **Attention back-tracking** method for computing the importance score of each input token by their importance on the required output tokens (e.g.: sequence representation token, in last layer) and the task performance. As illustrated in Figure 1 (right), we take a backward pass from the last layers' token representations to input tokens, pruning the tokens by their importance score of each input token by accumulating its attention score. By doing so, we are able to better select and keep the important tokens that will minimally affect the output and the prediction. We name our novel token pruning method as **S**parse **T**oken **T**ransformer with **A**ttention **B**ack-**T**racking (**STTABT**).

However, one challenge here is that such backtracking requires us to know the attention score of each token before feed-forwarding the input sequence to the model. To handle the issue, we introduce a lightweight attention approximation network trained with knowledge distillation called **ApproxNet**. Moreover, to actually prune the tokens, we need to decide which tokens to be retained based on the importance scores. The top-k method, by design, can select a predefined number of tokens (Goyal et al., 2020), but requires sorting the tokens by their importance scores, which is not differentiable. The thresholding method, which prunes the tokens with scores under the threshold value (Kim et al., 2022) does not require sorting, but setting the threshold can be tricky since it is difficult to know how many tokens will be remained in advance with the given threshold. To remedy such issues in both methods, we propose a learnable and smooth threshold function named **Concrete masking**, inspired by Concrete Dropout (Gal et al., 2017). Specifically, we jointly train the threshold value for each layer and importance score which is computed by attention back-tracking, with the task objective to find the thresholds that can minimize the task loss.

Our method is generally applicable to Transformer encoders for any domains, and we validate our method on text and image classification tasks. For text classification, we validated the proposed token pruning methods on GLUE (Wang et al., 2019) benchmark with BERT (Devlin et al., 2019). For image classification, we validate it on ImageNet-1K (Deng et al., 2009) benchmark with DeiT (Touvron et al., 2021). The experimental results show that our model works surprisingly well even at very high pruning rates, for which baselines show significant performance degeneration. For example, on a GLUE benchmark task (QQP), our method obtains 45.54% increased token sparsity (17.15% to 9.34%) compared to the baseline with the same accuracy. Moreover, our method obtains very high token sparsity (18.7%), keeping only half of the tokens retained by DynamicViT (37.5%), with only 0.8% accuracy loss on ImageNet-1K benchmark. Our contributions can be summarized as follows:

- We propose a novel token pruning method for Transformers based on attention back-tracking, which considers the importance of each token on the final representations as well as the task loss based on the approximated attention from the distilled model.
- We propose Concrete masking, which automatically learns the pruning rate for each layer with Concrete dropout. (Gal et al., 2017).

- Our token pruning is generally applicable to any Transformers, and we validated it on trained Transformers from both NLP (BERT) and CV (DeiT) domains, on GLUE and ImageNet benchmark, whose results show that it achieves a significantly better accuracy-computation tradeoff. (Kim et al., 2022; Rao et al., 2021).

## 2 RELATED WORK

**Attention Pruning for Transformers** Since the literature on the general neural network compression techniques such as quantization and weight pruning is vast, we discuss compression techniques specifically targeting Transformers (Vaswani et al., 2017). Due to the heavy computational cost of the self-attention, several recent works have focused to reduce its time complexity from $\mathcal{O}(N^2)$ to about $\mathcal{O}(N)$ given the $N$ length of the sequence by pruning the attentions in the self-attention layers. BigBird (Zaheer et al., 2020) is one such method, and it utilizes set of sparse attentions, that are both local and random, while using few global attentions, to deal with longer sequences. Similarly, Star-Transformer (Guo et al., 2019) leverages the concept of a virtual relay node and reduces the number of fully-connected attentions with the radial and ring connections. However, these prior works do not consider the content of the input sequence, as they make use of static sparsity patterns. To remedy this, some recent works aim to dynamically sparsify attentions of the Transformer considering the input data. Specifically, Reformer (Kitaev et al., 2020) divides the tokens into several groups through Locally Sensitive Hashing (LSH) and calculates attentions by sparsely connecting them. Routing Transformer (Roy et al., 2021) uses the k-means clustering algorithm to cluster input tokens.

**Token Pruning for Transformers** However, in most datasets used in Natural Language Processing (NLP), such as text classification (Wang et al., 2019), input sequences with with excessively long lengths are quite rare, as the length of most sequences is limited to 256 tokens (Kim et al., 2022). Moreover, fully connected layers require significant computations that are non-negligible, although they have been relatively overlooked. Therefore, some recent works (Kim et al., 2022; Goyal et al., 2020) focus on reducing the number of tokens by selecting the most important tokens for the given task, to reduce the computational cost of both linear and self-attention layers. Notably, PoWER-BERT (Goyal et al., 2020) progressively eliminates tokens based on their importance scores, computed from the attentions at each layer of the Transformer encoder, and Learned Token Pruning (LTP) (Kim et al., 2022) aims to learn to threshold the importance scores to prune the token. However, both methods prune the tokens during the feed-forward pass, assuming that important input tokens are also important output tokens. Therefore, they may prune out tokens that are important for the output vector at earlier layers. Compared to them, our method STTABT can consider the impact of each token on the output vector and the prediction task by attention back-tracking, thanks to our novel lightweight attention approximation network.

## 3 SPARSE TOKEN TRANSFORMER WITH ATTENTION BACK TRACKING

We start by briefly describing the Transformer model that our method focuses on. Then, we propose two components for our Sparse Token Transformer with Attention Back Tracking (STTABT): *Attention Back Tracking (ABT)* which obtains the importance score for each token with an attention approximation network and *Concrete masking* for dynamically learning the token-wise threshold for each input. STTABT is a post-hoc token pruning method and assumes that the target Transformer encoder has previously been optimized on the training dataset for the target task. Our goal is to lower the inference cost of the given pretrained Transformer by pruning the tokens at each layer.

### 3.1 TRANSFORMER

For concepts used in later sections, we define the self-attention layer — the fundamental element of the Transformer architecture (Vaswani et al., 2017) — in this section. Let us denote the attention score and probability at $l$-th Transformer layer as follows:

$$\boldsymbol{A}^l_{score} = \frac{\boldsymbol{Q}^l \boldsymbol{K}^{l^\top}}{\sqrt{d_k}}, \qquad \boldsymbol{A}^l_{prob} = \text{softmax}(\boldsymbol{A}^l_{score}), \tag{1}$$

where $\boldsymbol{Q}^l \in \mathbb{R}^{L \times d_k}$ is the query matrix, $\boldsymbol{K}^l \in \mathbb{R}^{L \times d_k}$ is the key matrix, $d_k$ is the dimension of query and key matrix, and $l \in [1, L]$ is the layer index.

## 3.2 ATTENTION BACK-TRACKING (ABT)

We now describe the details of our method, Attention Back Tracking (ABT). The fundamental idea of ABT is to prune the tokens based on their importance scores that measure their impacts on the final sequence representation for task prediction, by back-tracking the attentions. Specifically, as shown in Figure 1 (right), we start with the tokens with higher attention probabilities from the last layer, and backtrack their accumulated attention probabilities to the first layer of the Transformer encoder consisting of $L$ layers with regards to the token for the sequence classification task (specifically, [CLS] token). However, to apply such a technique into practice, we must feed-forward the input sequence through all of the Transformer layers in order to compute attention scores beforehand. Thus a straightforward implementation of the technique could be quite inefficient because it will require us to perform forward pass of the model twice for each input.

**Attention Approximation Network**   To alleviate the cost, we use an *attention approximation network* (ApproxNet), a distilled Transformer encoder that approximates the attention probabilities of the original network with a significantly less number of parameters. We perform knowledge distillation based on the distillation loss from TinyBERT (Jiao et al., 2020) to build the ApproxNet. Specifically, we leverage embedding based distillation loss $\mathcal{L}_{embd}$, hidden state based distillation loss $\mathcal{L}_{hidn}$, attention-based distillation loss $\mathcal{L}_{attn}$, and prediction-layer distillation loss $\mathcal{L}_{pred}$ for the distillation. Especially, for the attention-based distillation, we use the KL divergence loss from MiniLM (Wang et al., 2020) instead of the mean squared error loss as follows:

$$\mathcal{L}_{attn} = \frac{1}{HT^2L} \sum_{l=1}^{L} \sum_{h=1}^{H} \sum_{j=1}^{T} \sum_{i=1}^{T} (\boldsymbol{A}^l_{prob} * (\log \boldsymbol{A}^l_{prob} - \log \hat{\boldsymbol{A}}^l_{prob}))_{h,j,i}, \quad (2)$$

where $\boldsymbol{A}^l_{prob} \in \mathbb{R}^{H \times T \times T}$ denotes $l$-th layer attention probability of the original network (teacher), $\hat{\boldsymbol{A}}^l_{prob} \in \mathbb{R}^{H \times T \times T}$ denotes $l$-th layer attention probability of the ApproxNet (student), $H$ is the number of heads for multi-head self-attention layers, $T$ is the length of input sequence, and $L$ is the number of layers in the Transformer network.

**Algorithm 1:** Update token mask from the output token indices

```
1 def UpdateTokenMaskOfLayer(Â_prob^l, i^{l+1}, s^{l+1}, p, k)
     // Reduce attention heads and gather required
        output tokens' attention probabilities
        with previous importance score.
2    A ← gather(mean(Â_prob^l, dim=0), i^{l+1}, dim=0)
3    a ← mean(diag(s^{l+1})A * p + A * (1-p), dim=0)
4    i^l ← unique(cat(i^{l+1}, topk(a, k)))
5    s^l ← gather(a, i^l)
6    s^l ← s^l / sum(s^l)
7    return i^l, s^l
```

Figure 2: Computation flow of update token mask of whole Transformer encoder layers with attention back-tracking by Algorithm 1.

**Token Pruning with Attention Back-tracking on Approximated Attention**   Then, using the approximated attentions from the ApproxNet, we can now prune the tokens at each layer of the main network. In Algorithm 1, we define the pruning function UpdateTokenMaskOfLayer, which updates the mask at each Transformer layer $l$ for token pruning. The input arguments of the pruning function are the approximated attention probability $\hat{\boldsymbol{A}}^l_{prob}$ of $l$-th layer from the ApproxNet, the token mask $\boldsymbol{i}^{l+1} \in \mathbb{Z}^{o_{l+1}}$, and the importance scores of the tokens $\boldsymbol{s}^{l+1} \in \mathbb{R}^{o_{l+1}}$ from the upper layer $l+1$, where $l = 1$ is the first layer of the Transformer model and $o_{l+1}$ is the number of selected tokens in $(l+1)$-th layer. $\boldsymbol{i}^L$ is initialized with indices of required tokens, which is [CLS] token in the classification task. $\boldsymbol{s}^L \in \mathbb{R}^{o_T}$ is initialized as the one vector $\mathbf{1}/o_T$ where $o_t$ is the number of output

tokens, which is generally a single token. $p \in (0, 1) \subset \mathbb{R}$ determines how much importance scores should be propagated from the upper layer. If $p = 0$, the importance scores from the upper layer will be completely ignored when pruning the tokens in $l$-th layer. $k \in \mathbb{Z}$ determines how many tokens should be selected in case of using top-$k$ selection method, and we could easily change the number of tokens to retain at each layer, by adjusting $k$. The function returns the index of remaining input tokens at $l$-th layer and their importance scores given those arguments.

### 3.3 CONCRETE MASKING

The top-$k$ tokens at each layer of the Transformer model may still be selected using the previous scheme's static value of $k$, but in order to regulate the pruning based on the input, a mechanism for dynamically altering the pruning ratio is also necessary. Therefore, previous work, such as LTP (Kim et al., 2022) proposed to learn a layer-wise threshold for token pruning. In detail, during fine-tuning, LTP keeps tokens that have larger scores than the threshold, which is jointly optimized with the network parameters. However, it can be challenging to control the number of tokens left at each layer when the importance score is relatively sparse and the majority of score values are close to zero or very small values. To address the problem, we suggest a novel dynamic thresholding method named as *Concrete Masking*, inspired by the learnable dropout function idea in Concrete dropout (Gal et al., 2017).

Let us consider a single row of one of multi-head attention score matrix $\boldsymbol{a}_t^l = \boldsymbol{A}_{score}{}_{h,t}^l \in \mathbb{R}^T$ in this section for brevity, where $h$ is the head index and $i$ is the row index. We further assume that the attention scores follow a normal distribution. We transform $\boldsymbol{a}_t$ into the uniform distribution where $\boldsymbol{a}_{t,i} \in [0, 1]$ so that the importance score becomes uniformly distributed. Formally,

$$\tilde{\boldsymbol{a}}^l = \Phi(\frac{\boldsymbol{a}^l - \mu_{\boldsymbol{a}}}{\sigma_{\boldsymbol{a}}}), \qquad \tilde{\boldsymbol{A}}^l = \begin{bmatrix} - & \tilde{\boldsymbol{a}}_1^l & - \\ & \cdots & \\ - & \tilde{\boldsymbol{a}}_T^l & - \end{bmatrix} \tag{3}$$

where $\Phi$ is the cumulative distribution function of the standard normal distribution, $\mu_{\boldsymbol{a}} = \sum_{t=1}^T \boldsymbol{a}_t^l / T$ is the mean of $\boldsymbol{a}^l$, and $\sigma_{\boldsymbol{a}} = \sqrt{\sum_{t=1}^T (\boldsymbol{a}_t^l - \mu_{\boldsymbol{a}})^2 / T}$ is the standard deviation of the $\boldsymbol{a}^l$. Then, we apply above transformation to all the rows of the predicted attention score matrix from ApproxNet $\hat{\boldsymbol{A}}_{score}^l$ to acquire $\tilde{\boldsymbol{A}}_l$ where all of rows are uniformly distributed. Similar to the line 3 of Algorithm 1, we then propagate the transformed importance score $\tilde{\boldsymbol{a}}$ as follows:

$$\boldsymbol{b}^l = \frac{\sum_{t=1}^T \left( (p \mathrm{diag}(\boldsymbol{s}^{l+1}) + 1 - p) \mathrm{diag}(\boldsymbol{m}^{l+1}) \tilde{\boldsymbol{A}}^l \right)_{t,:}}{\sum \boldsymbol{m}^{l+1}}, \quad \boldsymbol{s}^l = \left( 1 - \epsilon \frac{\boldsymbol{b}^l}{\max \boldsymbol{b}^l} + \epsilon \right) \tag{4}$$

where the token mask $\boldsymbol{m}^{l+1} \in (0, 1)^T \subset \mathbb{R}^T$ is the Concrete mask vector. Then, the updated Concrete token mask $\boldsymbol{m}^t$ is defined as follows (Gal et al., 2017):

$$\boldsymbol{m}^l = \max \left( \boldsymbol{m}^{l+1}, \mathrm{sigmoid} \left( \frac{\log p_{\theta^l} - \log(1 - p_{\theta^l}) + \log \boldsymbol{s}^l - \log(1 - \boldsymbol{s}^l)}{\tau} \right) \right) \tag{5}$$

where *sigmoid* is the sigmoid function, $\tau$ is the temperature hyperparameter, and $p_{\theta^l} = \mathrm{sigmoid}(\theta^l)$ where $\theta^l$ is a learnable parameter and initialized with $\theta_{init}$. During training, the model learn the proper value of $\theta^l$ with the soft mask function applied to each layer. After training for reproducing the base accuracy with sufficient epochs, we change the soft mask into a hard mask, then fine-tune the parameters to fit the hard mask as in LTP (Kim et al., 2022). For hard masking, we threshold the soft Concrete token mask $\boldsymbol{m}^l$ using a set threshold, usually $0.5$.

To train our Concrete masking mechanism, we need regularization terms in addition to the task loss. First of all, we add the regularization term $\mathcal{L}_p$, which prevents $\theta^l$ from deviating too far from the initialized value and another regularization term $\mathcal{L}_{mask}$ so that the average token retention ratio stay around $p_{\theta_{init}}$ as follows:

$$\mathcal{L}_p = \lambda_p \sum_{l=1}^L (\theta^l - \theta_{init}^l)^2, \quad \mathcal{L}_{mask} = \lambda_{mask} \left( \left( \frac{1}{TL} \sum_{l=1}^L \sum_{t=1}^T \boldsymbol{m}_t^l \right) - p_{\theta_{init}} \right)^2 \tag{6}$$

where $\lambda_p$ and $\lambda_{mask}$ are hyperparameters.

# 4 EXPERIMENTS

## 4.1 EXPERIMENTAL SETUP

To show the general applicability of our method, STTABT, we validate it on both Natural Language Processing (NLP) and vision tasks. In order to evaluate our method, we primarily compare the performance of our method against those of the baselines at the same **token retention ratios**, which is a metric that measures how many tokens are retrained after pruning. We use the average token retention ratio values across all the layers of the target Transformers. To accurately measure the efficiency of our method, even considering the overhead for computing the ApproxNet for attention approximation, we also report the performance as a function of FLOPs in Section 5.

### 4.1.1 PRUNING TRANSFORMER-BASED TRAINED LANGUAGE MODELS

**Dataset and metric.** We use nine datasets from GLUE (Wang et al., 2019) benchmark for the text classification and use the BERT$_{base}$ (Devlin et al., 2019) as the base model. For evaluation metric, we use the metrics from the original paper (Wang et al., 2019). **Baselines.** We compare STTABT against two notable token pruning baselines – the manual top-k method and Learned Token Pruning (LTP) (Kim et al., 2022). The manual top-k method is the feed-forward token pruning which keeps the $k_l$ tokens with highest importance scores for each $l$-th layer. LTP is a baseline with a learnable threshold, which requires to fine-tune the whole parameters to learn the appropriate threshold value for each input. We match the training settings of STTABT and LTP, then follow the hyperparameter search strategies described in (Kim et al., 2022). For the comparison against LTP, we use STTABT with Concrete masking strategy (§3.3), while use top-k pruning with our method (§3.2) when comparing against the manual top-k baseline.

### 4.1.2 PRUNING VISION TRANSFORMERS

**Dataset and metric.** We use ImageNet-1k (Deng et al., 2009) for the image classification experiment, using the Data-efficient Image Transformer (DeiT) (Dosovitskiy et al., 2021a; Touvron et al., 2021) as the base Vision Transformer model. Specifically, we use DeiT-small as the base architecture for all experiments on the image classification. **Baselines.** We use DynamicViT (Rao et al., 2021) as the primary baseline. We follow the experimental setups, including data preprocessing and training pipeline of DynamicViT for all experiments on the image classification task.

## 4.2 EXPERIMENTAL RESULTS ON NLP TASKS

**Attention back-tracking.** We measure the performance of our attention back-tracking (ABT) method with the ApproxNet (§3.2). We experiment with different $k$ in Algorithm 1 to vary the average token retention ratios. The experimental results in Figure 3 show that the our method significantly outperforms the manual top-k with the same average token retention ratio. Furthermore, we find that using the ApproxNet (blue lines) obtains similar performance to using the attention from the full network (green lines). These results support that our attention backtracking strategy can identify more important tokens than the feed-forward pruning methods do, which compute the importance scores and perform pruning with forward passes.

**Concrete masking.** In Figure 3, we further present the performance evaluations with or without the Concrete masking, to examine its effect. One of the advantages of our model with Concrete masking is that the user can flexibly adjust the token retention ratio by changing $\theta_{init}$ value. In contrast, the user cannot easily adjust the token retention ratio of LTP since the method does not support the control of the token and relies on regularization on the mask, which is the token retention ratio with a regularization scaling factor. For the lower token retention ratio of LTP, we must increase the scaling factor, then the regularization term becomes significantly bigger than task loss at some point. As a result, the LTP token retention ratio converges to zero or an extremely low ratio which is not appropriate to deal with task loss. For instance, in QNLI, MRPC, CoLA, and SST-2 of Figure 3, LTP baseline cannot cover the low token retention ratio, whereas our method can.

At relatively high token retention ratios around 30-50%, both STTABT with concrete masking and LTP work well, although our method still outperforms LTP in most cases. However, at lower than 30% token retention ratios, our method with Concrete masking show significantly superior performance

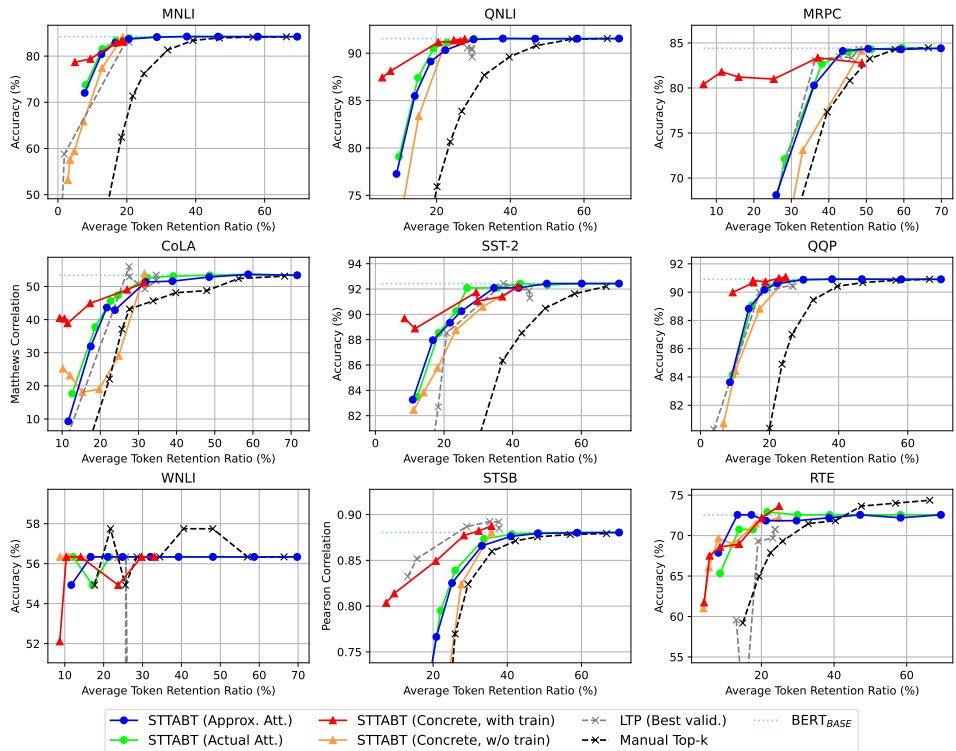

Figure 3: Experimental results with varying average token retention ratio for our method STTABT (by ApproxNet with factor 4) and baselines on GLUE and BERT$_{base}$. Our results with top-k pruning (blue, green) are compared against the manual top-k method (black dot). In contrast, results with Concrete masking (red, orange) are compared to the LTP (grey dot).

over LTP, better preserving the accuracy of the full model (Please see the results on MNLI, QNLI, MRPC, CoLA, SST-2, QQP, and RTE). To examine which tokens are pruned with our method and LTP, we further visualize the pruning mask at each layer of BERT$_{base}$ in Figure 6. We observe that LTP keeps most of the important tokens, but unimportant tokens as well such as dots in Figure 6. On the other hand, our method with Concrete masking prunes useless tokens such as dots at much earlier layers than LTP, yielding higher sparsity in total.

The Concrete masking method have significantly higher accuracy in most of cases in GLUE benchmark than ABT without Concrete masking and baselines. The Concrete masking method works especially well at high pruning rates, while have competitive performance to baselines at low pruning rates too.

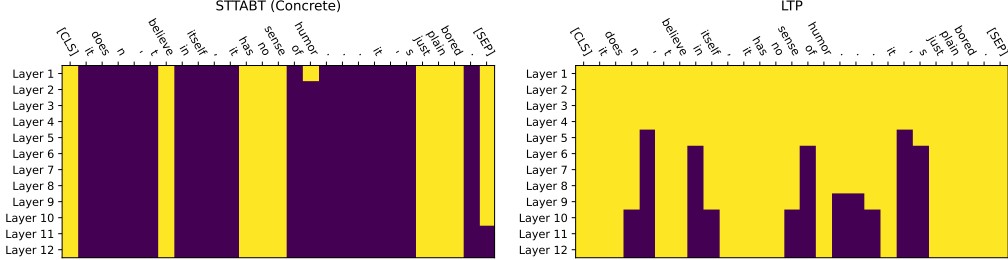

Figure 4: Token pruning visualization between our Concrete masking and LTP layer by layer. Layer $l$ means the input token mask on the layer $l$. Yellow color indicates the remained input tokens in each layer. The input texts are randomly selected from The Stanford Sentiment Treebank (SST-2) (Wang et al., 2019). Similar accuracy of trained LTP and our Concrete masking models are used ($\approx 89\%$).

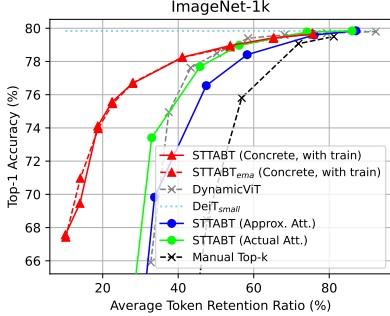

Figure 5: Top-1 accuracy plot varying the average token retention ratio for our method and baselines on ImageNet-1k.

Figure 6: The visualization of token pruning with STTABT (Concrete) and DynamicViT. Models with similar accuracy are used (Ours@f4: 74.1%, DynamicViT: 74.9%).

Table 2: Relative FLOPs of various overhead sources of our methods. The FLOPs results are calculated with a sequence length of 128. We randomly generate token retention ratios for each layer for simulation. Fused MulAdd for matmul is not considered in FLOPs calculation.

| | $BERT_{base}$ | ApproxNet | | | Manual Top-k | | | Attention Back-Tracking | | | Concrete Masking | | |
|---|---|---|---|---|---|---|---|---|---|---|---|---|---|
| | | Reduce Factor | | | Token Retention Ratio | | | Token Retention Ratio | | | Token Retention Ratio | | |
| | | 4 | 8 | 16 | 0.2 | 0.4 | 1.0 | 0.2 | 0.4 | 1.0 | 0.2 | 0.4 | 1.0 |
| FLOPs | 22.4G | 1.5G | 436.9M | 140.6M | 559.5K | 1.1M | 2.8M | 597.4K | 1.2M | 2.8M | 9.2M | 18.5M | 46.1M |
| Relative | 100% | 6.86% | 1.95% | 0.63% | 0.00% | 0.00% | 0.01% | 0.00% | 0.01% | 0.01% | 0.04% | 0.08% | 0.21% |

## 4.3 EXPERIMENTAL RESULTS ON COMPUTER VISION TASKS

The experimental results show that our Attention back-tracking (ABT) performs competitively to DynamicViT even without Concrete masking, on computer vision tasks, and our full model with Concrete masking significantly outperforms Dynamic ViT 5, especially at lower token retention ratios. For example, our method with Concrete masking achieves $74.1\%_{(-5.8\%p)}$ accuracy with $18.7\%$ token retention ratio, but DynamicViT requires $37.5\%$ token retention ratio with $74.9\%_{(-5.0\%p)}$ accuracy. Moreover, since our ABT requires training ApproxNet only once regardless of the target token retention ratio, it has a huge advantage in training cost than DynamicViT. The DynamicViT used twice more tokens than our Concrete masking, but the accuracy improvement is only $0.8\%$. Even considering attention approximation overhead, our model performs better than DynamicViT since the overhead is only about $1.95 - 6.86\%$ of the base model, as shown in Table 2.

We visualize the pruned tokens at different layers of the vision Transformers in Figure 6. The visualization shows that our model with Concrete masking concentrates on the important tokens in an earlier layer than DynamicViT. Moreover, our model preserves more important tokens while DynamicViT attends to a larger number of tokens, including unimportant ones. For instance, DynamicViT selects tokens with other attributes at Layer 4, while our model keeps only the tokens related to Chihuahua from the beginning of the model, thanks to the attention back-tracking in Figure 6.

## 5 COMPUTATIONAL EFFICIENCY OF STTABT

### 5.1 FLOPs COMPARISION

For the GLUE benchmark, we compare the attention back-tracking (ABT) method and the manual top-k method with respect to FLOPs. In Figure 7a, we compare the performance with respect to FLOPs over the test set. We find that the ABT performs better than the manual top-k baseline on most datasets.

Table 1: Approximation error of ApproxNet depending on initializing methods.

| | Init. | Random | | Distilled | |
|---|---|---|---|---|---|
| Dataset | Factor | 2 | 4 | 4 | 8 |
| Average | MSE | 0.0112 | 0.0112 | **0.0013** | 0.0018 |
| GLUE | KL Div. | 1.6430 | 1.6450 | **0.2019** | 0.2981 |

The ABT results look shifted toward the right than the Figure 3, because the computational cost includes overheads such as computing the mask update function (Algorithm 1) and the ApproxNet. We examine the relative overheads of each additional component in Table 2. We see that ApproxNet

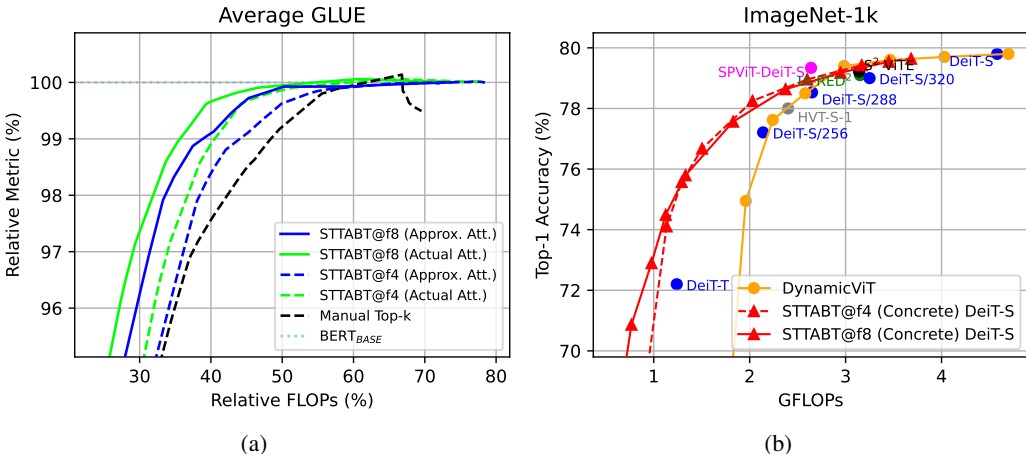

Figure 7: (a) FLOPs comparison between attention back-tracking with top-$k$ and manual top-$k$ baseline on BERT$_{base}$. (b) FLOPs comparison between our method with concrete masking and baselines on ViT. All plotted models except DeiT-T are based from DeiT$_{small}$ for fair comparison. STTABT@f$x$ means the attention back-tracking by ApproxNet with factor $x$. DeiT-S/$h$ means the DeiT$_{small}$ with hidden size $h$ (Kong et al., 2021).

is the major bottleneck, as the ApproxNet with factor 4 requires about $6.86\%$ of the BERT$_{base}$ FLOPs, which is significantly larger than the manual top-k ($0.01\%$). However, the overhead is more convincing with higher reduction factor of 8 ($1.95\%$) than 4. Therefore in Figure 7a, the ABT results with factor 8 shows better result than 4.

We further compare the performance of STTABT with different reduction factors of the ApproxNet. Table 1 shows that the ApproxNet performs accurately approximations of the actual attention scores with a lower reduction factor, thus results in better preformance. However, when considering the accuracy-efficiency tradeoff, the performance of the model with the reduction factor of 4 are often worse than the one with the reduction factor of 8.

In Figure 7b, we show the accuracy over FLOPs comparison of our model against baselines on ImageNet-1k (Deng et al., 2009), using DeiT$_{small}$ as the base model to prune. The baselines include DynamicViT (Rao et al., 2021) is used as the primary baseline, DeiT$_{small}$ (Touvron et al., 2021), SPViT (Kong et al., 2021), IA-RED$^2$ (Pan et al., 2021a), S$^2$ ViTE (Chen et al., 2021), and HVT (Pan et al., 2021b). We observe that the fine-tuned DeiT$_{small}$ model pruned by our Concrete masking significantly outperforms the fine-tuned DeiT$_{tiny}$ model with the same FLOPs trained from scratch. DynamicViT with DeiT$_{small}$, on the other hand, results in a model with much lower performance than DeiT$_{tiny}$. This is good news for practitioners, as this suggests that we may not need to train Transformers at different scales from scratch, to deploy the model to systems or applications with different computational budgets.

## 6 CONCLUSIONS

In this paper, we propose a sparse token Transformer architecture based on a novel token pruning method, Sparse Token Transformer with Attention Back Tracking (STTABT), with two novel ideas – Attention Back-Tracing (ABT) and Concrete masking. In particular, the ABT method enables our pruning method to back-track the important tokens for classification from the topmost layer to the lower layers, while approximating the attentions using a lightweight approximation network (ApproxNet). Then, we propose a novel thresholding method named as Concrete masking, which adapts the idea from Concrete dropout to learn the appropriate token-wise pruning threshold given the desired amount of tokens to be retained. We evaluate the performance of our method on classification tasks from NLP and CV domains, by pruning tokens from the trained Transformer encoder-based language models and Vision Transformers. Our method shows a better accuracy-efficiency trade-off and yields Transformers with significantly higher accuracy for the same ratio of tokens retained, or at the same FLOPs even when considering the computational overhead from the ApproxNet. Qualitative analysis of the tokens retained by STTABT shows that it preserves relatively more important tokens while pruning out less relevant ones, compared to previous methods.

## REPRODUCIBILITY STATEMENT

We will introduce code construction and data collection in this section for reproducibility.

**Model Implementation.** First we construct our model with PyTorch (Paszke et al., 2019) and Huggingface (Wolf et al., 2020). We modify `bert-base-uncased` from Huggingface Models (Wolf et al., 2020). **ApproxNet.** We create ApproxNet (§3.2, `trainer/glue_base.by`) with the pretrained models. The approximation result is shown in Section A. **ABT.** We implement Algorithm 1 on top of `bert-base-uncased` (Devlin et al., 2019) (`models/sparse_token.py`). **LTP, Concrete masking.** Then, we modify forward path to support LTP (Kim et al., 2022) and fix `ApproxSparseBertModel` to implement Concrete masking (§3.3).

**NLP training.** The training for NLP tasks, we implemented our own training loop (`trainer/concrete_trainer.py`) and hyper-parameter tuning settings inherits LTP hyper-parameter settings (Kim et al., 2022) (`main/ltp_glue_plot.py`, `main/concrete_glue_plot.py`). We use the GLUE dataset from Huggingface Dataset (Wolf et al., 2020).

**Vision training.** We start from DynamicViT implementation (Rao et al., 2021) including data loader, training loop, training loss, and evaluation codes (`main/vit_concrete_end2end.py`). Therefore, we did not apply hyper parameter grid search on ViT pipeline. And we remove hard token mask training stage, and we use larger $\lambda_{mask} = 100$ than NLP task, because of lack of padding. Also, we did not use prediction distillation loss for training ApproxNet for ViT. The training dataset is ImageNet-1k (ILSVRC2012) (Deng et al., 2009). We replace the Transformer encoder in ViT model (Dosovitskiy et al., 2021b) into our model code. Our PyTorch model is modified to apply ViT changes from Transformer encoder (Dosovitskiy et al., 2021b).

## ACKNOWLEDGMENTS

This work was supported by DeepAuto.ai. This work was partly supported by Institute of Information & communications Technology Planning & Evaluation (IITP) grant funded by the Korea government (MSIT) (No. RS-2022-00187238, Development of Large Korean Language Model Technology for Efficient Pre-training), (No.2022-0-00124, Development of Artificial Intelligence Technology for Self-Improving Competency-Aware Learning Capabilities), and (No.2019-0-00075, Artificial Intelligence Graduate School Program (KAIST)). This work was supported by the Engineering Research Center Program through the National Research Foundation of Korea (NRF) funded by the Korean Government MSIT (NRF-2018R1A5A1059921).

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

APPENDIX

# A  EXPERIMENTAL DETAILS

## A.1  DETAILS ON ATTENTION APPROXIMATION NETWORK

| | Init. | Random | | Interpolate | | Distilled | |
|---|---|---|---|---|---|---|---|
| **Dataset** | **Factor** | **2** | **4** | **2** | **4** | **4** | **8** |
| **MNLI** | **MSE** | 0.0087 | 0.0087 | 0.0086 | 0.0086 | **0.0013** | 0.0018 |
| | **KL Div.** | 1.7503 | 1.7520 | 1.7174 | 1.7097 | **0.2393** | 0.3423 |
| **QNLI** | **MSE** | 0.0052 | 0.0052 | 0.0052 | 0.0052 | **0.0009** | 0.0012 |
| | **KL Div.** | 1.8026 | 1.8078 | 1.7826 | 1.7750 | **0.2666** | 0.3866 |
| **MRPC** | **MSE** | 0.0052 | 0.0052 | 0.0052 | 0.0052 | **0.0007** | 0.0011 |
| | **KL Div.** | 1.9680 | 1.9751 | 1.9484 | 1.9406 | **0.2248** | 0.3740 |
| **CoLA** | **MSE** | 0.0339 | 0.0338 | 0.0337 | 0.0334 | **0.0032** | 0.0042 |
| | **KL Div.** | 1.0845 | 1.0810 | 1.0709 | 1.0629 | **0.1223** | 0.1644 |
| **SST-2** | **MSE** | 0.0167 | 0.0167 | 0.0166 | 0.0165 | **0.0019** | 0.0025 |
| | **KL Div.** | 1.4762 | 1.4744 | 1.4622 | 1.4528 | **0.1899** | 0.2606 |
| **QQP** | **MSE** | 0.0099 | 0.0099 | 0.0099 | 0.0098 | **0.0014** | 0.0020 |
| | **KL Div.** | 1.5960 | 1.6020 | 1.5756 | 1.5679 | **0.2269** | 0.3149 |
| **WNLI** | **MSE** | 0.0070 | 0.0070 | 0.0070 | 0.0069 | **0.0005** | 0.0009 |
| | **KL Div.** | 1.5982 | 1.5977 | 1.5752 | 1.5688 | **0.1375** | 0.2211 |
| **STSB** | **MSE** | 0.0097 | 0.0097 | 0.0096 | 0.0096 | **0.0009** | 0.0012 |
| | **KL Div.** | 1.5230 | 1.5266 | 1.4991 | 1.4915 | **0.1561** | 0.2258 |
| **RTE** | **MSE** | 0.0048 | 0.0048 | 0.0048 | 0.0048 | **0.0007** | 0.0010 |
| | **KL Div.** | 1.9883 | 1.9881 | 1.9678 | 1.9593 | **0.2533** | 0.3930 |
| **Average GLUE** | **MSE** | 0.0112 | 0.0112 | 0.0112 | 0.0111 | **0.0013** | 0.0018 |
| | **KL Div.** | 1.6430 | 1.6450 | 1.6221 | 1.6143 | **0.2019** | 0.2981 |

Table A.1: The ApproxNet performance results over GLUE benchmark with various initialization setting.

In this section, we provide additional details on the attention approximation networks (ApproxNet). We train the ApproxNet following the instructions in Section 3.2. As we mentioned, the architecture of the ApproxNet is identical with the original network such as BERT (Devlin et al., 2019) and Vision Transformer (ViT) (Touvron et al., 2021), except for the dimension of hidden and intermediate hidden unit size. We downsize the architecture by reducing the hidden unit dimension with different reducing factors. We use 4 and 8 as the reducing factors in all experiments of the paper. We then train the ApproxNet with the knowledge distillation (Wang et al., 2020; Jiao et al., 2020). Specifically, we use the embedding based distillation loss $\mathcal{L}_{embd}$, hidden state based distillation loss $\mathcal{L}_{hidn}$, attention-based distillation loss $\mathcal{L}_{attn}$, and prediction-layer distillation loss $\mathcal{L}_{pred}$ to train the ApproxNet for BERT. For Vision Transformer, we do not use the prediction-layer distillation loss.

In Table A.1, we present the approximation error between the attention probability in the original network and the ApproxNet. We observe that the distillation helps the ApproxNet to approximate almost identical attention probability of the original network. We also observe that the smaller reduce factor leads to lower attention approximation error. However, we also find that the ApproxNet with a larger reducing factor also accurately approximate the attention probability of the original network.

To boost the training speed for ApproxNet on downstream tasks, we first distill the pre-trained Transformer encoders prior to the distillation of the fine-tuned network on each task. This procedure is not necessary if the downstream task is large enough or there is no downstream task such as ImageNet-1k. For instance, we directly train ApproxNet by setting the fine-tuned ViT as the teacher network with ImageNet-1k dataset. However, for experiments on GLUE benchmark, we first train the ApproxNet by setting the original pre-trained BERT as the teacher network with the Wikitext-103 dataset (Merity et al., 2017) for 200 epochs. Then, we finetune the ApproxNet by setting the fine-tuned BERT on each task as the teacher network with each task dataset for 30~50 epochs.

# B   ADDITIONAL EXPERIMENTAL RESULTS

Figure B.1: FLOPs comparison between attention back-tracking with top-$k$ and manual top-$k$ baseline on all of datasets in GLUE benchmark. `STTABT@fx` means the attention back-tracking by ApproxNet with factor $x$.

## B.1   ADDITIONAL RESULTS ON GLUE EXPERIMENTS WITH FLOPS

In Figure B.1, we present evaluation results on all GLUE benchmark with regards to FLOPs. We average every data point across 9 datasets in Figure B.1 to plot Figure 7a in the main paper.

## B.2   ADDITIONAL VISUALIZATION EXAMPLES

In Figure B.2 and B.3, we visualize the additional visualization of the token pruning on each layer with STTABT (Concrete) and baselines (Kim et al., 2022; Rao et al., 2021). We further confirm that our method prunes more tokens throughout all of transformer layers than baselines in both sentence and image classification tasks.

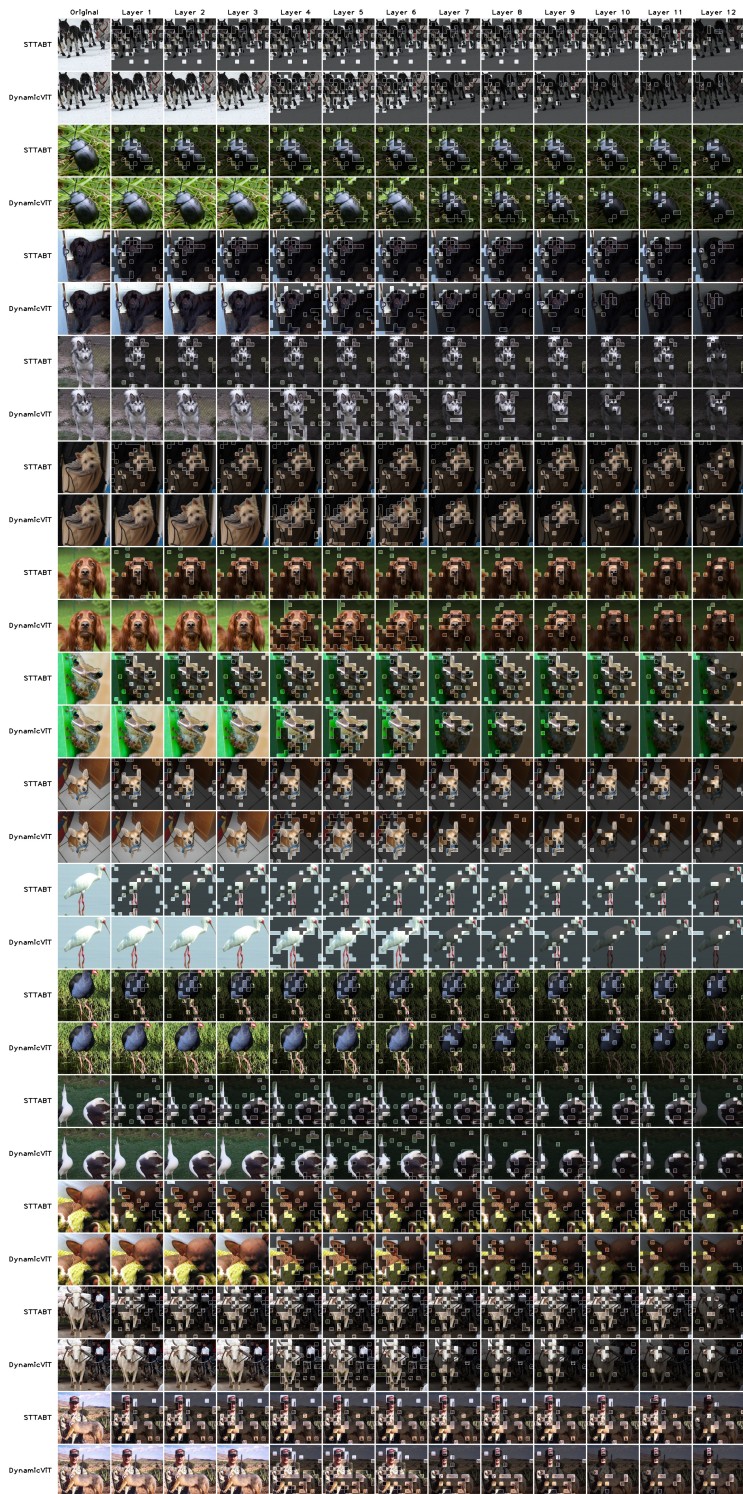

Figure B.2: More visualization examples of the token pruning on each layer with STTABT (Concrete) and DynamicViT on ImageNet-1k dataset.

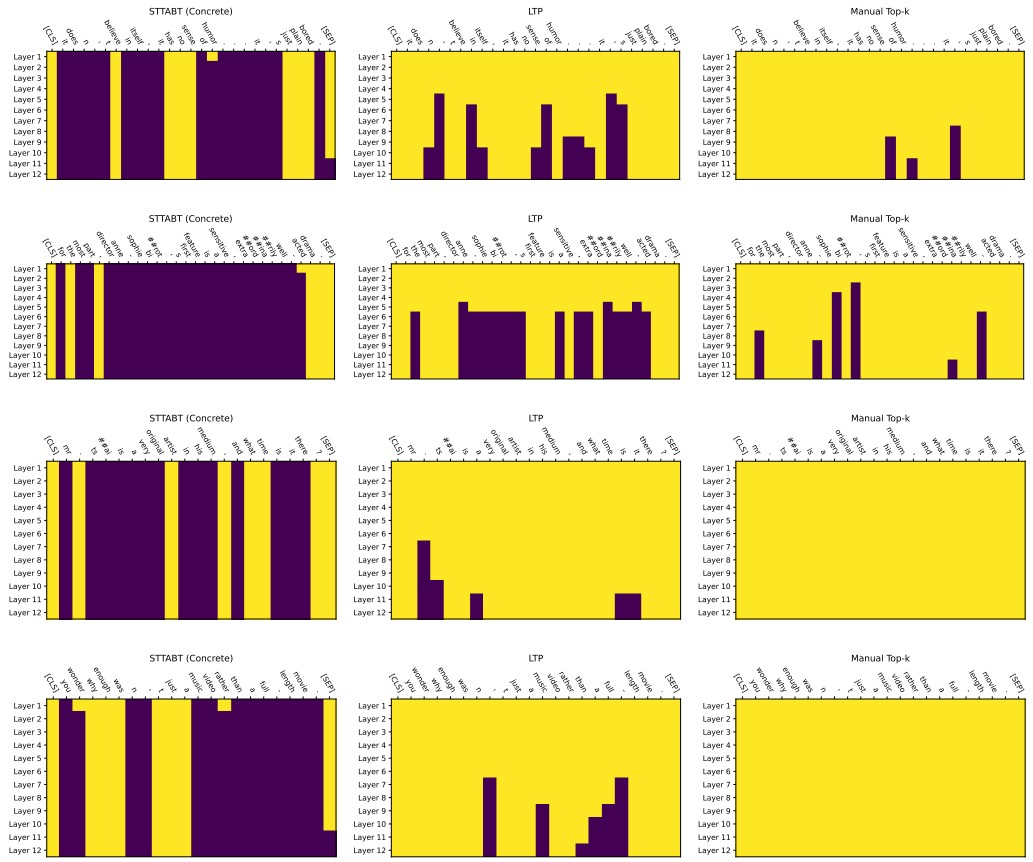

Figure B.3: More visualization examples of the token pruning on each layer with STTABT (Concrete), LTP, and Manual Top-k on SST-2 dataset.

# C    ABLATION STUDY

## C.1    PROPAGATION FACTOR $p$

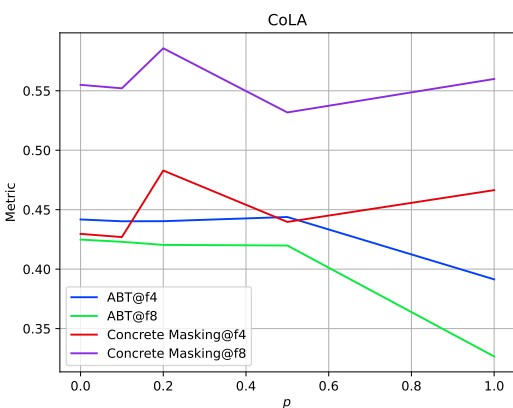

Figure C.4: Ablation study about propagation factor $p$ in Algorithm 1.

We perform an ablation study on the importance score propagation factor $p$, which is introduced in Algorithm 1. We use CoLA dataset in GLUE benchmark (Wang et al., 2019) for this ablation study.

In Figure C.4, we observe that Attention Back-Tracking method with $p$ value between the range of 0.0-0.5 shows the similar task performance. However, we also observe that using $p$ value as 0.1 performs the best.

## C.2  CONCRETE $\lambda_{mask}$ AND $\lambda_p$

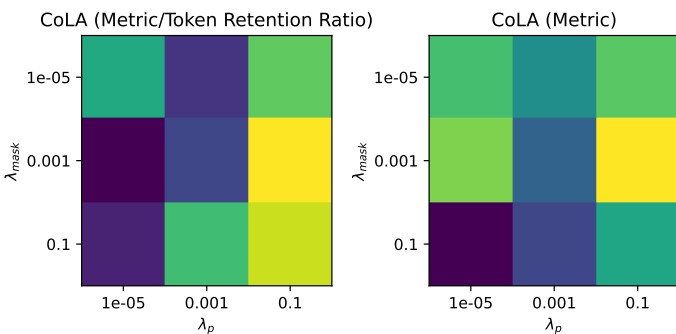

Figure C.5: Ablation study on $\lambda_{mask}$, $\lambda_p$ used in Concrete masking training (§3.3). Yellow indicates a relatively large value while dark blue indicates a relatively small value. (Left) Heatmap of task performance divided by token retention ratio. (Right) Heatmap of task performance.

We perform an ablation study on two balancing terms $\lambda_{mask}$, $\lambda_p$ used in Concrete masking training, which is introduced in Section 3.3. We use CoLA dataset in GLUE benchmark (Wang et al., 2019) for this ablation study. In Figure C.5, we could not observe any clear pattern from varying $\lambda_p$ and $\lambda_{mask}$. Therefore, it is better to adjust both values individually depending on the task. For instance, for tasks in GLUE benchmark, we observe $\theta^l$ does not change a lot during the training. Therefore, we use a low $\lambda_p$ around $1e-3$ in GLUE task training. On the other hand, we observe the model with concrete masking tend to fail in keeping the target token retention ration in the image classification task with ViT. In this case, we use a high $\lambda_{mask}$ around 100 to keep the token retention ratio close to the desired target value.

# D  EXTRA BASELINES ON VIT EXPERIMENT

## D.1  TRADE-OFF BETWEEN TOP-1 ACCURACY AND TOKEN RETENTION RATIO

Technically, it is possible to apply our proposed method on any transformer-based architecture. To validate this, we perform additional experiments on the image classification task with another ViT architecture (Jiang et al., 2021). In Figure D.6, we additionally plot results of STTABT (Concrete masking) on LVViT (Jiang et al., 2021). Please note that Figure D.6 is the extended version of Figure 5 where we add experimental results on STTABT (Concrete) with LVViT and the original LVViT (Jiang et al., 2021). Concrete masking on LVViT results in accuracy drop of $3.68\%p$ when we drop a token retention ratio from $76\%$ to $18\%$, while Concrete masking on DeiT suffers accuracy drop of $5.53\%p$ in the same range of token retention ratio. This result shows our Concrete masking method preserves the performance of the original model by preserving important tokens better.

## D.2  TRADE-OFF BETWEEN FLOPS AND TOKEN RETENTION RATIO

We additionally plot the results of STTABT (Concrete masking) with $LVViT_{small}$ (Jiang et al., 2021) and DynamicViT (Rao et al., 2021) with $LVViT_{small}$ in Figure D.7. Figure D.7 is an extended version of Figure 7b. When we use LVViT as the backbone model, Concrete masking outperforms DynamicViT. The additional baselines include MViTv2 (Li et al., 2022), SwinT (Liu et al., 2021), and LVViT (Jiang et al., 2021).

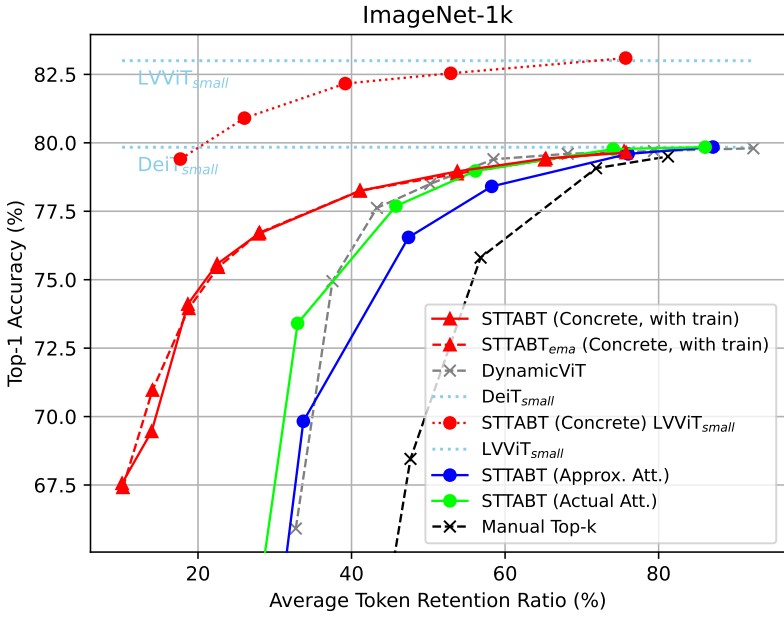

Figure D.6: Top-1 accuracy plot with results of STTABT (Concrete) with LVViT.

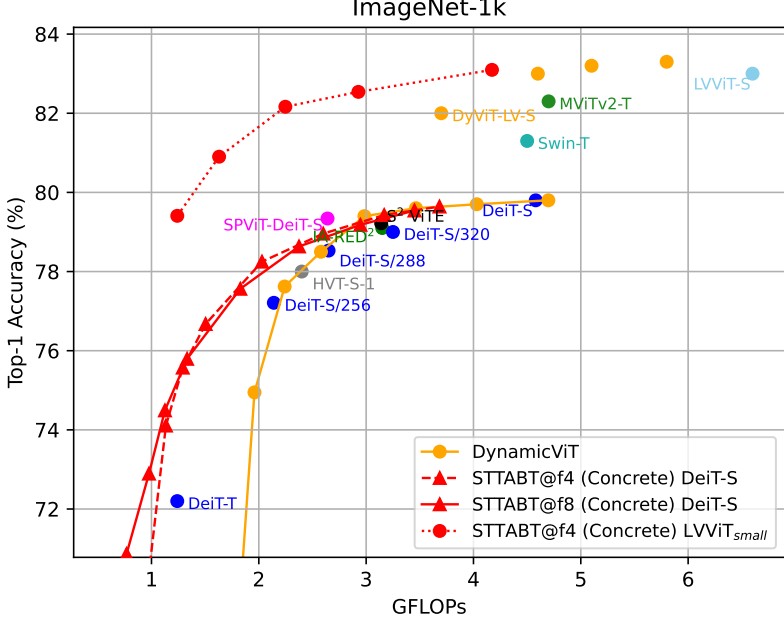

Figure D.7: Results of STTABT and DynamicViT using LVViT$_{small}$ as the backbone network. `DyViT-LV-S` means DynamicViT with LVViT$_{small}$. Note that this figure is the extended version of Figure 7b.

