# OpenReview forum: "Sparse Token Transformer with Attention Back Tracking"
_ICLR.cc/2023/Conference — ICLR 2023 poster_

### Official Review · Reviewer_qjDB · 2022-10-25

**Confidence:** 3
**Correctness:** 3
**Technical Novelty And Significance:** 3
**Empirical Novelty And Significance:** 2
**Recommendation:** 6

**Clarity, Quality, Novelty And Reproducibility:**

 The idea makes sense but the clarity of the method needs to be greatly improved. Given the current form, it would be difficult to reproduce the presented method.


**Strength And Weaknesses:**

The idea of attention backtracking makes sense as it is possible that some tokens are seemingly not important at the current layer but may be important at later layers. The proposed method also demonstrates good performance compared to some baseline methods.

I have some concerns about the clarity of the method and empirical evaluation.

How is ApproxNet trained? Is it trained along with the main Transformer model or separately trained after the main model training is done?

What does recursive pruning mean in Figure 1? I understand the main idea of the caption but find the figure hard to interpret.

What does DeiT-S/256, DeiT-S/288 mean in Figure 7(b)? Are 256/288 represent different resolutions? If so, why do they have lower accuracy than the original DeiT-S, which runs at resolution 224x224?

I found it’s a bit misleading to mostly use token retention ratio as the main metric for computation due to the additional cost of ApproxNet. It would be great to report FLOPs consistently for all methods and indicate how much is consumed by ApproxNet numerically.

How does the proposed method perform compared to other types of Vision Transformers, e.g., Swin Transformer or Multiscale Vision Transformers? If the proposed method is generic enough, it would be great to demonstrate results on other types of transformers too.


**Summary Of The Paper:**

This paper focuses on token pruning for Transformer models. The authors observe that previous token pruning approaches do not consider the impact of a token on later layers’ attentions. Therefore, they propose an attention back-tracking method that tracks the importance of each attention from the outputs to the inputs, based on which they can preserve tokens with a large impact on the final predictions. The method includes an attention approximation network (ApproxNet) that learns to approximate the attention probabilities and a Concrete masking mechanism to learn the thresholds for token pruning. The authors provided results on ImageNet and GLUE to validate the proposed method.


**Summary Of The Review:**

The idea is novel and makes sense to me. My concern is mainly about the clarity and evaluation of the method (reporting FLOPs numerically, more baselines and types of ViT architectures). I vote for borderline rejection and would like to see the rebuttal to make the final decision.

---

> ### Author Response · Authors · 2022-11-10
> **Our Answer to reviewer qjDB (2/2)**
>
> ---
>
> > **[Q2-4]** "How does the proposed method perform compared to other types of Vision Transformers, e.g., Swin Transformer or Multiscale Vision Transformers? If the proposed method is generic enough, it would be great to demonstrate results on other types of transformers too."
> - **Yes, we can definitely apply STTABT to Swin Transformer (SwinT) and Multiscale Vision Transformer (MViT) as well because STTABT is a generic algorithm and is thus applicable to any Transformers, including SwinT and MViT.** However, the current implementation does not support SwinT because it has a different architecture in the Attention layer (it uses multi-scale). Thus, it may take some time to implement STTABT for SwinT. Specifically, we have to modify all Concrete masking update functions for SwinT, which does not have a constant token length and constant attention connection shape. However, we are working on the implementation and include the results of our STTABT on SwinT for the final version of the paper.
> - Note that we already compared our method to **other types of Vision Transformers** in Figure 7b, such as **SPViT** [1], which uses token merging techniques. We showed that we could achieve very similar performance to SPViT without using additional techniques, such as token merging.
> - However, following the reviewer's suggestion, **we applied STTABT on another ViT architecture, such as LVViT** [2]. We reported corresponding experimental results in Figures D.6 and D.7 in the Appendix of the updated revision.
>
> ---
>
> > **[Q3]** Reviewer's concerns about reproducibility. "The idea makes sense, but the clarity of the method needs to be greatly improved. Given the current form, it would not be easy to reproduce the presented method."
> - **Please note that we provided all source codes of the model, training script, and hyper-parameter tuning scripts as the supplementary material.** Thus, we believe that anybody can quickly reproduce our experiment results, including all our plots in the paper.
> - For example, for the ViT experiment, the reproduction step is straightforward. All ApproxNet and Concrete Masking training are packed in a single python script. (All detailed usage description is in`README.md` files in the source code)
> ```bash
> python -m main.vit_concrete_end2end --n-gpus $NGPU --imagenet-root /path/to/ILSVRC2012/
> ```
> - We also hope that the supplementary file helps address your remaining concerns on reproducibility, as we provide all the necessary details to reproduce all experimental results there.
>
> [1] Zhenglun et al., SPViT: Enabling Faster Vision Transformers via Soft Token Pruning, ECCV 2022. [https://arxiv.org/abs/2112.13890](https://arxiv.org/abs/2112.13890)
>
> [2] Zihang et al., All Tokens Matter: Token Labeling for Training Better Vision Transformers, NeurIPS 2021. [https://arxiv.org/abs/2104.10858](https://arxiv.org/abs/2104.10858)

---

> > ### Comment · Reviewer_qjDB · 2022-11-21
> > **Concerns resolved**
> >
> > Thanks to the authors for their efforts during the rebuttal. My concerns have been resolved after considering the additional results and clarification in the response. Therefore, I decided to upgrade the rating. I also suggest the authors revise the paper to reflect their work in the rebuttal. Thanks.

---

> > > ### Author Response · Authors · 2022-11-21
> > > **Thank you for response**
> > >
> > > Thank you for your response. We are glad that our response successfully resolved your concerns. As you suggested, we uploaded the revised version of the paper which includes the Appendix with more ablation studies and a baseline comparison with LVViT.
> > >
> > > Please let us know if you have any additional questions or suggestions, as we will be more than happy to address them. Thank you.

---

> ### Author Response · Authors · 2022-11-18
> **Our Answer to reviewer qjDB (1/2)**
>
> Dear, Reviewer qjDB.
>
> Thank you for your considerable efforts in reviewing our paper and insightful reviews of our work. We believe that addressing your feedback will help us further improve the quality of our paper. We faithfully addressed most of your concerns and requests in our paper - the updated parts of the texts are denoted with green color. We address the individual comments in your review below:
>
> ---
>
> > **[Q1]** Reviewer's concerns about Figure 1, "What does recursive pruning mean in Figure 1?"
> - We used the term "recursive pruning" since the same pruning algorithm is applied to the next layer in a recursive manner (higher layer in the feed-forward case / lower layer in attention backtracking case). We could rename this process as "progressive pruning" if this sounds more intuitive.
> - A detailed illustration of such a progressive pruning scheme is in Figure 2. We hope that this illustration will help you better understand our algorithm.
>
> ---
>
> > **[Q2]** Reviewer's concerns about the clarity of the method and empirical evaluation.
> >
> > **[Q2-1]** "How is ApproxNet trained? Is it trained along with the main Transformer model or separately trained after the main model training is done?"
> - We described the training of ApproxNet as a separate knowledge distillation (KD) step (In Section 3.2 Attention Back-Tracking (ABT), the second paragraph, "Attention Approximation Network").
> - **Yes, ApproxNet will be separately trained after the main model training is done**. Please note that the training of ApproxNet is done just once per task (main model). Since ApproxNet is tiny and trained only once, the total training time of any number of pruned networks is less than or similar to DynamicViT, and the efficiency gain will become larger when training multiple pruned networks.
>
> ---
>
> > **[Q2-2]** “What do DeiT-S/256 and DeiT-S/288 mean in Figure 7(b)? Are 256/288 represent different resolutions? If so, why do they have lower accuracy than the original DeiT-S, which runs at resolution 224x224?"
> - Please note that 256 and 288 **denote the hidden unit size** of the modified DeiT-S, which initially has a hidden unit size of 384, and not the image resolution. Since the hidden unit size of DeiT-S is reduced, its accuracy becomes lower. All our experiments are done with 224x224 resolution images from the ImageNet 1k dataset.
> - We included the descriptions about the hidden unit size of DeiT-S in the revision. Thank you for addressing this.
>
> ---
>
> > **[Q2-3]** "I found it's a bit misleading to mostly use the token retention ratio as the main metric for computation due to the additional cost of ApproxNet. It would be great to report FLOPs consistently for all methods and indicate how much is consumed by ApproxNet numerically."
> - We believe that reporting how the accuracy varies with the token retention ratio is extremely important in verifying our main hypothesis that STTABT will select better "tokens" for the given classification task with a reduced number of computed tokens than the feed-forward method. This is why we used the token retention ratio as our metric for comparison in Section 4.
> - **Also, please note that we already report the overall numerical efficiency (FLOPs) of STTABT against multiple baselines in a separate section, in Section 5, Computational Efficiency of STTABT.**
> - However, following your suggestion, we also included the accuracy of STTABT and baselines as a function of FLOPs in Figure D.7. **The results show that STTABT largely outperforms all baselines in FLOPs as well, and the trend remains the same.**

---

### Official Review · Reviewer_ibLQ · 2022-10-25

**Confidence:** 2
**Correctness:** 3
**Technical Novelty And Significance:** 3
**Empirical Novelty And Significance:** 3
**Recommendation:** 6

**Clarity, Quality, Novelty And Reproducibility:**

Clarity:
The experiments section is well written. However, the method section (subsection 3.2 and subsection 3.3) may need some modifications to further improve the readability.

Quality:
The paper proposes a novel backtracking pruning method and validates it in both NLP and CV tasks.

Novelty:
The proposed method is new in the token pruning literature.

Reproducibility:
Good reproducibility. The authors include a reproducibility statement section and the code is attached for review. I checked the major part of the codes and it matches the proposed method in the main paper.

**Strength And Weaknesses:**

Strength:
1. The idea of using approximated network during the token pruning seems new in literature.
2. Promising accuracy and computation trade-off in the low token retention ratio regime.

Weaknesses:
1. The readability of the paper can be improved. For example, the right part of Figure 1 gives too much detailed information. Without referring to the contents of the later sections, it could be hard to serve as an illustration picture in the introduction section. Figure 2 is also a little bit confusing.
2. The proposed method requires a reliable lightweight distilled counterpart (ApproxNet) of the model to prune. When ApproxNet is not a good approximation of the original model, the proposed method may fail.


**Summary Of The Paper:**

This paper studies a novel token pruning/sparsification technique via backtracking the importance from the final layer to the first layer. In order to control the computation cost, the token importance is approximated via a forward procedure through a lightweight distilled counterpart (i.e., ApproxNet), and then a smoothed threshold function (i.e., Concrete masking) is adaptively learned to balance the sparsity level and total importance to preserve. The authors verify the efficiency of the proposed method on both GLUE with BERT base and ImageNet-1k classification with DeiT.

**Summary Of The Review:**

This paper considers a sparse token Transformer architecture based on a novel token pruning technique. The proposed technique contains two components: attention backtracing with an approximated model and concrete masking via a learnable threshold function. The authors validate the efficiency of the proposed method in both CV and NLP tasks. In particular, for the low token retention ratio regime, the proposed method beat the benchmarks by a significant margin.

Minor issues:

1. Influence of the ApproxNet. In the proposed method, the token importance is approximated via the ApproxNet. Thus the quality ApproxNet may play a critical role. I suggest authors add a section to directly discuss the influence of the accuracy of ApproxNet.

2. The proposed method still uses Top-k selection, which breaks the linear computation cost. I'm wondering if we can remove it or replace it with some other thresholding policies to maintain the linear cost.


I currently tend to accept this paper according to the method's novelty and good numerical performance and I'm willing to change my evaluation after the rebuttal.

---

> ### Author Response · Authors · 2022-11-18
> **Our Answer to reviewer ibLQ**
>
> Dear, Reviewer ibLQ.
>
> Thank you for your considerable efforts in reviewing our paper, and we are thankful for your detailed review of our work, including the code review for reproducibility. We believe that addressing your feedback will help us further improve the quality of our paper.
> We updated our paper by addressing the concerns and reflecting the suggestions from all reviewers, including yours. The updated parts are denoted with green color.
>
> ---
>
> > **[Q1]** Our correction to the reviewer's claim. "The proposed method still uses Top-k selection, which breaks the linear computation cost. I'm wondering if we can remove it or replace it with some other thresholding policies to maintain the linear cost."
> - Thank you for pointing out the computational cost of top-k selection. Please note that we are aware of the limitations of the top-k selection, that it requires sorting operation and is not differentiable. However, it is just a variant of our method, and our final token pruning method is based on a novel **Concrete masking (Section 3.3)** that **does not require top-k selection or sorting**. Concrete masking transforms our ABT methodology into a **smooth threshold learning problem** while keeping the ordinal information of the importance score of each token. Concrete masking achieves superior performance and efficiency over top-k ABT and Manual top-k thresholding methods.
> - To clarify, our token pruning framework is comprised of two novel methods, **Attention Back-Tracking (ABT; Section 3.2)** and **Concrete masking (Section 3.3)**. ABT uses attention back-tracking for calculating the importance score of each token and dropout tokens by selecting topk important tokens. Concrete masking uses a learnable smooth threshold concrete masking function for thresholding important tokens.
> - You can find the results of the Concrete masking method in Figure 5 and the efficiency of the Concrete masking method, including all overheads, in Figure 7b.
>
> ---
>
> > **[Q2]** General concern about readability "The readability of the paper can be improved.", "However, the method section (subsection 3.2 and subsection 3.3) may need some modifications to further improve the readability."
> - We understand your concern and agree that Figure 1 may look quite busy. However, we believe that the information in the illustration of Figure 1 is important in understanding our methodology.
> - The illustration on the right side of Figure 1 contains our key ideas, such as:
>     1. Attention Back-tracking (with reclusive pruning)
>         - How we can select important tokens in inputs of each layer (Topk vs. Learnable threshold)
>         - How we apply the importance score of output tokens to calculate the important score for input tokens layer by layer.
>     2. ApproxNet
>     3. Task independence
>         - Our method directly applies to both CV and NLP tasks.
>
> ---
>
> > **[Q3]** Reviewer concern about the potential reliability of ApproxNet "The proposed method requires a reliable lightweight distilled counterpart (ApproxNet) of the model to prune. When ApproxNet is not a good approximation of the original model, the proposed method may fail."
> - We also presumed that ApproxNet might have a huge impact on performance. However, in practice, ApproxNet works quite well even if we reduce the main network by factor 8 in Table A.1 and Figure 7a/b. Although the performance of ApproxNet with factor 8 is 1.69 times worse in KL divergence score performance (e.g., `WNLI 0.1375@factor4` → `0.2211@factor8`, from Table A.1), the purpose of increasing the reduction factor is for efficiency. In terms of efficiency, we showed a reduction factor of 8 might be more efficient than a reduction factor of 4 in Figure 7a. However, they both showed similar efficiency in the ViT experiment in Figure 7b, which means we may reduce ApproxNet by more than the reduction factor of 8.
> - In conclusion, ApproxNet generally works well in various ranges of reduce factors with regard to accuracy. However, in terms of efficiency, the appropriate reduction factor may be larger.

---

> > ### Author Response · Authors · 2022-11-25
> > **Dear Reviewer ibLQ,**
> >
> > Dear Reviewer ibLQ,
> >
> > We kindly notify you that the end of the discussion stage is approaching.
> >
> > Please check our response to your review. If you have any additional questions or concerns after reading our response, please let us know. We will address them as soon as possible.
> >
> > We thank you again for your time and efforts in reviewing our paper, as well as your constructive comments.
> > We are looking forward to your response.
> >
> > Best regards, Authors

---

> > > ### Comment · Reviewer_ibLQ · 2022-12-06
> > > **Thanks for responses.**
> > >
> > > Dear Authors,
> > >
> > > Thanks for your responses. They have alleviated many of the concerns voiced by me. In my opinion, I can not quite improve the score to an "8", but I would have given the paper a "7" if I could.

---

> > > > ### Author Response · Authors · 2022-12-07
> > > > **Thanks for your responses.**
> > > >
> > > > Thank you for your responses.
> > > >
> > > > We are glad to hear that we could alleviate your concerns and that the reviewer wants to increase the score to "7" if he/she could.
> > > >
> > > > Please let us know if you have any additional questions or suggestions, as we will be more than happy to address them. Thank you.

---

### Official Review · Reviewer_yJGa · 2022-10-26

**Confidence:** 4
**Clarity, Quality, Novelty And Reproducibility:** The paper is clearly written, and the…
**Correctness:** 4
**Technical Novelty And Significance:** 3
**Empirical Novelty And Significance:** 3
**Recommendation:** 6

**Strength And Weaknesses:**

In terms of strengths, the problem tackled by the paper is both relevant and important these days. Also, the paper provides experiments both for language and vision models, something I really appreciate. It also contains a number of interesting ablations.

In terms of weaknesses, the proposed algorithm is quite convoluted and it can't be applied directly during training. After training a model, it involves further training of new components (ApproxNet) that will be required during inference. I think this will most likely hinder its use in real-world practical setups.

A few questions:

- What's the size for VIT? is it Base?

- What's the impact of hyper parameter p (defined at the end of page 4)? Can we see performance as a function of p?

- At least for Computer Vision, there are very competitive algorithms [1, 2] that merge tokens (rather than dropping) for Transformers. Accordingly, they are end-to-end differentiable --so no top-K involved anywhere-- and the cost savings are applied and realized also during training (which leads to massive overall savings). It seems both can reduce the number of tokens to just 8 (even from say 256) in the middle of the network, thus matching performance while saving 40-50% of the training and inference costs. It would be nice to discuss the merits of merging (probably way more practical nowadays) and somewhat compare performance wrt these type of algorithms.

[1] = Learning to Merge Tokens in Vision Transformers, Renggli et al.
[2] = TokenLearner: Adaptive Space-Time Tokenization for Videos, Ryoo et al.

**Summary Of The Paper:**

The paper proposes an algorithm to remove a number of tokens during inference passes for Transformer models. The goal is to save time with respect to a full pass (i.e. no token dropping) while sacrificing as little performance as possible. The solution contains two main ingredients: an attention approximation network and the application of concrete masking.

The core idea is that we should be able to identify the important tokens to keep by looking at the attention scores from the whole network, as they suggest how much some tokens affect others. In particular, the paper claims it is essential to start from the latest layers and recursively propagate the scores backwards. Accordingly, methods that operate on a single forward pass can't identify the right tokens (as they "must" make decisions at layer L before seeing the scores of subsequent layers L+1, L+2, ...). The proposal for dealing with this is to train a small auxiliary network that --given a pre-trained "main" network-- learns to predict the intermediate attention scores for every layer directly from the inputs via distillation. This way, given a new input at inference, we can apply this network (ApproxNet) first, and use these predictions as proxy to do the token dropping.

The second idea tackles the problem of thresholding. How many tokens should we drop? A simple approach is to set some fixed value K, and always keep K. I guess this can be easily extended to K_i for layer i (as long as there's some monotonicity). The paper claims it may be hard to set K as it may be suboptimal --and probably K* is input dependent-- so they propose a method that drops all tokens whose score is below some learnable threshold per layer.

The paper then presents an extensive set of experiments, both for language and vision models.

Figure 3 shows that using STTABT with ApproxNet outperforms forward pass manual top-K, suggesting the former is able to better pick tokens. Figure 3 also shows that approximating the true "future" attention values with the ApproxNet doesn't lead to almost any performance loss wrt the real attention values. Concrete masking seems to be helpful when keeping very few tokens. However, this figure does not reflect the overhead of running ApproxNet. In other words, simpler forward-pass approaches can process more tokens at the same cost as STTABT processing fewer. Figure 7 tries to highlight this fact --while it uses FLOPs rather than time.

Figure 4 is cool and informative but maybe you could add the manual top-k choices too; Figure 6 is a bit hard to see (too small).

Figure 5 shows STTABT beats DynamicVIT for pretty much every retention ratio.

**Summary Of The Review:**

This paper proposes a way to reduce the number of processed tokens at inference by a pre-trained Transformer model. It trains a small network to predict attention scores, and learns a per-layer threshold to decide which and how many tokens to drop. A number of language and vision experiments suggest the method works well especially when we want to keep very few tokens. Alternative approaches that tend to be very efficient (like merging) aren't mentioned or discussed.

While the algorithm is a bit intricate, I think there's value in showing that using attention scores to select tokens can indeed work well.

---

> ### Author Response · Authors · 2022-11-18
> **Our Answer to reviewer yJGa**
>
> Dear, Reviewer yJGa.
>
> Thank you for your considerable efforts in reviewing our paper and insightful reviews. We believe that addressing your feedback will help us improve the quality of our paper even more. We address your individual comments below:
>
> ---
>
> > **[Q1]** Response to the question about ViT size "What's the size for VIT? is it Base?"
> - Thank you for pointing it out. We use **DeiT-small** for all experiments and baselines, for which the input size is fixed as 224x224. We did mention it in the caption of Figure 7b, however, we have added additional descriptions on the size of VITs in Section 4.1.2 in the revised version of the paper for better clarity.
>
> ---
>
> > **[Q2]** Response to the question about p "What's the impact of hyperparameter p (defined at the end of page 4)? Can we see performance as a function of p?"
> - We appreciate your suggestion on intriguing analysis. To address your question, we have conducted additional experiments with varying $p$ during the rebuttal period.
> - **We have included experimental results on varying $p$ in Section C.1 of the revised version of the paper.** We observe that a small $p$ value between $0.1$ and $0.3$ is suitable for both ABT with top-k and concrete masking. These findings suggest that the model marginally benefits from the importance score from the upper layer $\boldsymbol{A}^{l+1}$ in order to compute the current layer's importance score $\boldsymbol{A}^l$ more accurately.
>
> ---
>
> > **[Q3]** "Figure 4 is cool and informative, but maybe you could add the manual top-k choices too."
> - Thank you for your suggestion. We also believe the token pruning visualization of the manual top-k method in Figure 4 can be helpful. The main reason we did not add the visualization of the pruned tokens for the manual top-k method is because it rarely drops the token when achieving a similar accuracy of 89% to ours in the SST-2 dataset, which renders the visualization less meaningful. **However, to address your concern, we included the token pruning visualization of the manual top-k method for comparison in Appendix Figure B.3 of the updated revision.**
>
> ---
>
> > **[Q4]** At least for Computer Vision, there are very competitive algorithms [1,2] that merge tokens (rather than dropping) for Transformers. Accordingly, they are end-to-end differentiable --so no top-K involved anywhere-- and the cost savings are applied and also realized during training (which leads to massive overall savings). It seems both can reduce the number of tokens to just 8 (even from say 256) in the middle of the network, thus matching performance while saving 40-50% of the training and inference costs. It would be nice to discuss the merits of merging (probably way more practical nowadays) and somewhat compare performance wrt these types of algorithms.
> - Thank you for bringing out the important point. We agree with your opinion that token merging is also an essential direction, at least in the Computer Vision field.
> - However, we want to emphasize that our proposed token pruning is **orthogonal** and **not directly comparable** to the previous token merging methods due to the following differences:
>     1. Existing token merging methods [1,2] require a massive scale of pre-training from scratch since they should modify the backbone architecture. In contrast, our STTABT can be easily and efficiently applicable to pre-trained models **without re-training**.
>     2. Our proposed method is a generic approach that is applicable to any Transformer encoder architectures, and we validated it on **two different modalities (images and texts)**. This is a key difference since previous token merging methods [1,2] are only applicable to the image domain.
> - However, we find your insight as very helpful, and believe that it will be possible to combine our proposed token pruning method with the token merging method to further reduce computational costs, thanks to the orthogonality of both methods.
>
> [1] Renggli et al., Learning to merge Tokens in Vision Transformers, arxiv 2022. [https://arxiv.org/abs/2202.12015](https://arxiv.org/abs/2202.12015)
>
> [2] Ryoo et al., TokenLearner: Adaptive Space-Time Tokenization for Videos, NeurIPS 2021. [NeurIPS proceding link](https://proceedings.neurips.cc/paper/2021/file/6a30e32e56fce5cf381895dfe6ca7b6f-Paper.pdf)

---

> > ### Comment · Reviewer_yJGa · 2022-11-21
> > **Thanks for the clarifying comments**
> >
> > I'd like to thank the authors for their answers and comments to my questions, and their updates on the paper.

---

> > > ### Author Response · Authors · 2022-11-22
> > > **Thank you for your response**
> > >
> > > Thank you for your response. We are glad that our response successfully answered your questions. As you suggested, we uploaded the revised version of the paper, which includes the Appendix with more ablation studies and a manual top-k visualization.
> > >
> > > Please let us know if you have any additional questions or suggestions, as we will be more than happy to address them. Thank you.

---

### Official Review · Reviewer_7U3T · 2022-10-29

**Confidence:** 4
**Correctness:** 3
**Technical Novelty And Significance:** 3
**Empirical Novelty And Significance:** 2
**Recommendation:** 8

**Clarity, Quality, Novelty And Reproducibility:**

This paper is clearly written and easy to follow. For example, Algorithm 1 and all equations are helpful in understanding the exact mechanism of attention back tracking and Concrete Masking. In my view, figures are somewhat complicated. Making it clearer might be better.

The flexibility in choosing initial values is good. On the other hand, I presume it may increase the complexity of hyperparameter space. The authors should provide the detail on how they set those values and how difficult or costly to decide them.


**Strength And Weaknesses:**

STTABT shows better efficiency-accuracy compared to other token pruning baselines. The paper is well-motivated. STTABT is successful in two modalities, NLP and CV, indicating its generality to any modality that uses a transformer.

Although attention back tracking utilizes information from the later layers, it is somewhat heuristic and also induces additional costs (forward pass of ApproxNet and additional computations). The authors should explain these costs.

In terms of computational efficiency, latency instead of FLOPs might be more important for practitioners.

An ablation study and intrinsic evaluation of each component (ABT and Concrete Masking) are necessary.

Although the authors provided representative token pruning methods (PoWER-BERT, LTP, and DynamicVIT), there are other relevant works that might be more advanced than the included ones in NLP and CV. It would be great if a more comprehensive literature survey and comparison with them were included.


**Summary Of The Paper:**

This paper proposes a novel token pruning method based on attention back tracking for efficient transformer inference. The goal (and the difference from the previous token pruning methods) is to mitigate the mistakes of removing important tokens at lower layers. An attention approximation network (ApproxNet) is trained by distillation objectives and the attention probability is used to calculate token masks and significance scores at all transformer layers. The authors also apply Concrete Masking for dynamic thresholding. They have shown the effectiveness of STTABT on NLP and CV datasets, outperforming previous methods having higher sparsity while keeping the accuracy.

**Summary Of The Review:**

I enjoyed the paper because the proposed methods are well supported by reasonable motivations (e.g., token pruning decision based on the final prediction and learnable dynamic thresholding). Moreover, STTABT achieves good performance, meaning the practical usefulness of this paper.

---

> ### Author Response · Authors · 2022-11-18
> **Our Answer to reviewer 7U3T**
>
> Dear, Reviewer 7U3T.
>
> Thank you for your considerable efforts in reviewing our paper. We sincerely thank you for your constructive and helpful comments, and  believe that addressing your feedback will help us further improve the quality of our paper.  We address all your concerns below:
>
> ---
>
> > **[Q1]** Reviewer’s concern about additional costs “(forward pass of ApproxNet and additional computations). The authors should explain these costs."
> - We respectfully agree with your concern about the overhead of ApproxNet. As you pointed out, the cost for ApproxNet is not measured with the token retention ratio metric. However, please note that **we do report the overall numerical efficiency **(FLOPs) of STTABT against multiple baselines in Section 5**, Computational Efficiency of STTABT, and Figure 7.**
> - In addition, we also report the overhead of each individual component in Table 2. The results show that the computational costs used in ApproxNet become marginal as we increase the reduce factor.
> - The reason we use the token retention ratio as the metric for comparison in Section 4, is because we believe that reporting **how the accuracy varies with the token retention ratio is extremely important** in verifying our main hypothesis that STTABT will select better "tokens" for the given classification task with a reduced number of computed tokens than baselines.
> ---
>
> > **[Q2]** Reviewer's concern about latency "In terms of computational efficiency, latency instead of FLOPs might be more important for practitioners."
>
> - Thank you for the insightful suggestion. We also agree that latency is an important metric to measure the practical efficiency of the proposed method.
> - However, unfortunately, this is out of our work's scope. In this work, we majorly focus on developing the method that **appropriately prunes the tokens** by their importance score in the Transformer encoder with the Attention Back-tracking and the Concrete masking. To this end, we reported the accuracy while varying with the token retention ratio and the numerical computational efficiency (FLOPs) in experiments to backup our main hypothesis that **STTABT will select better "tokens" for the given classification task with a reduced number of computed tokens** than the baselines.
> - To implement our method with lower latency, **we need an optimized kernel because the existing library (e.g., PyTorch) does not provide a sparse channel linear layer from out of the box.** Specifically, we have to remove the masked token with `Tensor.gather` and `Tensor.scatter` to implement the sparse token linear and attention layer. However, in this case, the latency will not be much improved compared to the naive linear layer since such functions require duplicated memory copies. To overcome this issue, we should customize CUDA kernel. Therefore, in this work, we multiply binary masks with hidden units when we need the sparse token, for ease of development. We will implement an optimized sparse token layer for both the linear and attention layer, and release the code as soon as the implementation is done.
>
> ---
>
> > **[Q3]** Reviewer's concern about the ablation study of components "An ablation study and intrinsic evaluation of each component (ABT and Concrete Masking) are necessary."
> - Thank you for pointing it out. We also believe more analysis of our method might be helpful for practitioners. Therefore, we have conducted additional ablation studies on propagation factor $p$ and balancing terms $\lambda_p, \lambda_{mask}$ for concrete masking. **We included results and discussions on ablation studies in Appendix C.1 and C.2 of the updated revision.**
>
> ---
>
> > **[Q4]** Reviewer's concern about the literature survey and comparison "there are other relevant works that might be more advanced than the included ones in NLP and CV. It would be great if a more comprehensive literature survey and comparison with them were included."
> - Thank you for your suggestion on the additional comparison. First of all, at least in the CV domain, we have compared our method against **various state-of-art baselines in Figure 7** and provided literature comparisons in Sections 4 and 5.
> - However, to fully address your concern, we included additional comparisons using other ViT architectures as the backbone model to validate the flexibility of our proposed method. Specifically, **we have added experimental results** on the image classification task with LVViT [1] in Appendix Section D in the updated revision.
>
> [1] Jiang et al., All Tokens Matter: Token Labeling for Training Better Vision Transformers, NeurIPS 2021.  [https://arxiv.org/abs/2104.10858](https://arxiv.org/abs/2104.10858)

---

> > ### Author Response · Authors · 2022-11-25
> > **Dear Reviewer 7U3T,**
> >
> > Dear Reviewer 7U3T,
> >
> > We kindly notify you that the end of the discussion stage is approaching.
> >
> > Please check our response to your review. If you have any additional questions or concerns after reading our response, please let us know. We will address them as soon as possible.
> >
> > We thank you again for your time and efforts in reviewing our paper, as well as your constructive comments.
> > We are looking forward to your response.
> >
> > Best regards, Authors

---

> > > ### Comment · Reviewer_7U3T · 2022-11-25
> > > **Thanks for the response**
> > >
> > > Dear authors,
> > >
> > > Thanks for the author response.
> > > I've read your answer, and it partially addresses my concerns.
> > > Anyway, I keep my current score.
> > >
> > > Thanks,
> > > Reviewer 7U3T

---

### Author Response · Authors · 2022-11-18
**General Response**

# General Response

We sincerely appreciate all the reviewers for their insightful comments. We are delighted that all the reviewers acknowledge the technical novelty of our method. We have responded to the individual reviews below and believe that we have faithfully responded to almost all of them. We have also included most of the suggestions in the revision.
Here we briefly summarize the updates we have made to the revision:

- We have added missing descriptions about DeiT-S variants and fixed the wrong references for the SPViT paper.
- We have added ablation studies on propagation factor $p$ in Appendix.
- We have added ablation studies on hyperparameter $\lambda_p$ and $\lambda_{mask}$ for concrete masking in Appendix.
- We have added the STTABT evaluation result with LVViT in Appendix.
- We have added the token mask visualization of the manual top-k baseline in Appendix.

We highlight the updated part of the revision as the green text.

Below, we summarize additional experimental results during the rebuttal period:

### Ablation: Propagation factor p (Section C.1)

- We could observe a small difference in p in the range of $0.0$-$0.5$ on ABT. However, p in the range of $0.1$~$0.3$ works well on the Concrete Masking algorithm. We found that our choice of $p=0.1$ is a reasonable value of $p$ because large $p$ performs worse, and $p$ with a smaller value performs worse with Concrete masking.

We included this result in the Appendix. **Please check the revised version of the paper for the plots.**

### Ablation: Concrete $\lambda_p$ and $\lambda_{mask}$ (Section C.2)

- We could not observe a clear pattern on $\lambda_p$ and $\lambda_{mask}$. Thus, we suggest keeping both lambdas with small values, such as $1e-3$, and increasing them depending on the task.
- We concluded that $\lambda_p$ does not need to be changed because, in most GLUE subsets, $\theta^l$ does not change a lot during the training step.
- $\lambda_{mask}$ should be changed if the trained concrete model does not keep the target token retention ratio. In the ViT experiment, we used $100$ for $\lambda_{mask}$ to make the token retention ratio learned by Concrete masking close to the target token retention ratio.

We included this result in the Appendix. **Please check the revised version of the paper for the plots.**

### Evaluation of STTABT on LVViT (Section D)

- We evaluated our STTABT (Concrete masking) on LVViT and included results from LVViT, SwinT, and MultiScaleViTv2 papers.
- Concrete masking on LVViT loses accuracy $3.68$%p from a token retention ratio $76$% to $18$%, while Concrete masking on DeiT loses $5.53$%p in the same token retention ratio range.
- Concrete masking with LVVIT outperforms DynamicViT with LVViT.

We added these results in the Appendix with additional baselines. **Please check the revised version of the paper for the plots.**

---

### Decision · Program_Chairs · 2023-01-20

**Decision:**

Accept: poster

**Justification For Why Not Higher Score:**

While the paper is interesting and I recommend acceptance, I don't see enough novelty in the proposed method nor potential impact to justify a spotlight or oral presentation.

**Justification For Why Not Lower Score:**

This is a valuable paper that seems of interest to the research community.

**Metareview: Summary, Strengths And Weaknesses:**

This paper proposes an algorithm to prune tokens at inference time in transformer models. Two main ingredients are proposed: an attention approximation network (ApproxNet) and concrete masking. The authors addressed the main concerns of the reviewers in their rebuttal and in the updated manuscript. All reviewers feel positively about this paper, and so do I. I therefore recommend acceptance.



**Note From Pc:**

if the above contains the word "oral" or "spotlight" please see: "oral" presentation means -> notable-top-5% and "spotlight" means -> notable-top-25%. As stated in our emails, we are disassociating presentation type from AC recommendations